# Targeted Attacks on Timeseries Forecasting

## Abstract

Real-world deep learning models developed for Time Series Forecasting are used in several critical applications ranging from medical devices to the security domain. Many previous works have shown how deep learning models are prone to adversarial attacks and studied their vulnerabilities. However, the vulnerabilities of time series models for forecasting due to adversarial inputs are not extensively explored. While attack on a forecasting model might aim to to deteriorate the performance of the model, it is more effective, if the attack is focused on a specific impact on the model's output. In this paper, we propose a novel formulation of Directional, Amplitudinal, and Temporal targeted adversarial attacks on time series forecasting models. These targeted attacks create a specific impact on the amplitude and direction of thre output prediction. We use the existing adversarial attack techniques from the computer vision domain and adapt them for time series. Additionally, we propose a modified version of the Auto Projected Gradient Descent attack for targeted attacks. We examine the impact of the proposed targeted attacks verses untargeted attacks. We use KS-Tests to statistically demonstrate the impact of the attack. Our experimental results demonstrate how targeted attacks on time series models are viable and are more powerful in terms of statistical similarity. It is, hence difficult to detect through statistical methods. We believe that this work opens a new paradigm in the time series forecasting domain and represents an important consideration for developing better defenses.

## 1 Introduction

Time Series Forecasting (TSF) tasks are seen in many real-world problems across several domains. The wide range of domains of applications include demand forecasting Carbonneau et al. (2008), anomaly detection Laptev et al. (2017), stock price prediction jae Kim (2003), electrical pricing Crespo Cuaresma et al. (2004) and weather forecasting Grover et al. (2015). Improved availability of data and computation resources has reflected in the recent efforts (Rasul et al. (2020), Wen et al. (2017), Oreshkin et al. (2019)) of applying deep learning techniques for forecasting tasks. The wide applications of such deep learning models have led to threat due to adversaries and hence also the work towards exploration and prevention (Rathore et al. (2021), Cao & Gong (2017), Li & Li (2020)) of such adversarial attacks.

For a given classification model, the goal of the adversary could be either targeted or untargeted. In targeted attacks, the adversary tries to misguide the model to a particular class other than the true class. In an untargeted attack, the adversary tries to misguide the model to predict any of the incorrect classes. The definition of targeted are well-defined for classification tasks and have been used in several previous works (Goodfellow et al. (2014a), Kurakin et al. (2016), Croce & Hein (2020)). These definitions are not applicable for regression tasks such as TSF. In the adversarial machine learning domain, time series tasks have received significantly less attention as compared to those of computer vision. Also, the adversarial attacks and defenses studied in the computer vision domain are not always useful for time series, requiring specific adaptations and re-definitions.

In this paper, we address the above-mentioned shortcomings by providing a formulation for targeted attacks on TSF. To do this, we extend the popularly known adversarial attacks from the computer vision domain to time series forecasting. Together with popular attacks such as Fast Gradient Sign Method (FGSM) and Projected Gradient Descent (PGD), we also propose a modified variant of Auto PGD attacks. We perform KS-tests on the loss attributes of the output forecasts (predictions) to study the statistical properties of the proposed targeted attacks.

The contributions of our work are as follows:

1. We define and formalize targeted attacks on deep learning time series forecasting.

2. We propose a modified Auto PGD attack for Time Series Forecasting (mAPGD-TSF), an extension of the AutoPGD algorithm for targeted time series attacks, which can be extended to any regression task.

3. Through statistical tests, we show that inputs with targeted perturbations are much more indistinguishable than untargeted attacks through empirical studies on the Google Stock and Household electric power consumption datasets.

## 2 RELATED WORK

Most of the approaches on adversarial attacks were first started on image classification in the deep learning domain. Szegedy et al. (2013) proposed adversarial examples for image recognition, which initiated a direction to investigate adversarial attacks in various domains. Goodfellow et al. (2014b) proposed the Fast Gradient Sign Method (FGSM) which is a single-step attack. In a similar line, Kurakin et al. (2018) presented an iterative version of FGSM called the Basic Iterative Method (BIM). These attacks pave the way for the current state-of-art adversarial attacks in computer vision. Khamaiseh et al. (2022) summarizes the existing white-box and black-box adversarial attacks and compares them on the basis of time taken, strength, and transferability of the attacks. Additionally, by focusing on how they are employed in real-world applications, they have offered a thorough description of the most popular defense mechanisms against adversarial attacks.

Oregi et al. (2018) introduced the first adversarial attacks on Time Series Classification (TSC). Fawaz et al. (2019) used existing adversarial attack mechanisms such as the FGSM, BIM to decrease the accuracy of residual networks for Time Series Classification. Rathore et al. (2021) brings the first notion of targeted attack on time series data. The targeted attacks are however focused on TSC tasks. Yang et al. (2022) brings the notion of a black-box method called TSadv on the TSC task. This work suggests a gradient-free black-box strategy to attack DNNs for TSC with local perturbations based on time series shapelets and differential evolution, which is in contrast to prior work that required gradient information and global perturbations.

Pialla et al. (2022) introduced Smooth Gradient Method (SGM) attack based on a gradient method and shows how adversarial training is a good way to improve a time series classifier's (TSC) robustness against smoothed perturbations by enforcing a smoothness condition on generated perturbations that contains spike and sawtooth patterns. The work Karim et al. (2019), takes into consideration that the time series models are sensitive to abnormal perturbations in the input and stringent requirements on perturbations. To address this, the work crafts time series adversarial based on the importance of measurement. The adversarial inputs are subjected to models performing time series prediction tasks such as LSTNet, CNN-, RNN- and MHANET-based models. An importance-based adversarial attack needs much smaller perturbations compared to other existing adversarial attacks. The work, however, does not formulate or address the targeted attacks in time series forecasting.

Another work Mode & Hoque (2020) on time series forecasting, explores the vulnerabilities of deep learning multi-time series regression models to adversarial samples. The work also focuses on gradient-based white box attacks on deep learning models such as CNNs, Gated-Recurrent Units (GRU), and Long-Short Term Memory (LSTM) models. The vulnerabilities are shown to be transferable and have the ultimate consequences. This work also focuses on untargeted adversarial attacks with an aim to increase the error of the deep learning model's output. Dang-Nhu et al. (2020) considers an adversarial setting in a probabilistic framework on auto-regressive forecasting models. This work uses Monte-Carlo estimation in approximating the gradient of expectation and addresses the challenge of effectively differentiating through Monte-Carlo estimation using reparametrization and score-function estimators. This also proposes an under-estimation attack and an over-estimation attack on electricity consumption prediction for reparametrization and score-function estimators.

## 3 SETTINGS AND FORMULATIONS

### 3.1 THREAT MODEL

**Time Series Forecasting.** Given a time series $X = [x_1, x_2, ..x_T]$, a time series forecasting task predicts the value of $x_{T+1}$ based on the previous samples $[x_{T-w}, x_{T-w+1}, ...x_T]$, where $w$ is the window size under consideration. The sample $x_{T+1}$ corresponds to the forecasted value and is often represented by $\hat{Y}$.

**Adversarial Time Series**. An adversarial perturbation $\eta$, typically superposed on a given input time series $X$, to construct $\hat{X}$ given by $[\hat{x_1}, \hat{x_2}, ..., \hat{x_T}]$. The adversarial time series $\hat{X}$ ($X_{adv}$) is intended to significantly worsen the output prediction $\hat{Y}$ of a TSF model.

**Goal of Adversary** The goal of the attacker is to create a targeted output impact on the time series. We consider $L_\infty$-bounded perturbation that causes a targeted attack. We consider the definition of white-box access, where the gradients of the model are available for loss calculation. The Threat model can also be extended in a transfer attack scenario, where an Oracle model created through extraction in a black-box setting can be used as white-box. Further, the range of the input after the perturbation is assumed to be accepted by the model. The inputs are otherwise, limited to the acceptable input range of the model. We denote the regressor $f : \mathbb{R}^{(M)} \mapsto \mathbb{R}^{(N)}$ with parameters $\theta$, the predicted output for input $x \in \mathbb{R}^{(M)}$ is represented as $y = f(x)$.

**Properties of the perturbation.** In general, adding a perturbation to the input should detoriate the performance of the model prediction. Additionally, It is also important for the perturbation to satisfy additional requirements including: 1. Small changes to the input to create bigger performance degradation on the output. Larger perturbations to the inputs are also easily detectable by the input plausibility check modules. It is notable that it is more expensive to achieve higher input perturbation. 2. Imperceptible perturbations, attributing to reduced risk due to detection of the input perturbation. The perturbation is hence formulated as a constrained optimization problem, where $f$ is the Deep TSF model under consideration and $\epsilon$ indicates the strength of the attack,

$$\eta = maximize||Y - \hat{Y}||$$
$$\text{s.t } ||X - \hat{X}|| \le \epsilon, \text{ where } Y = f(X) \text{ and } \hat{Y} = f(\hat{X}) \tag{1}$$

### 3.2 KS TESTS IN TIMESERIES

Kolmogorov-Smirnov Test (KS Test) Massey (1951) is a popular non-parametric test method in statistics, to test whether a sample follows a reference probability distribution (one sample KS-Test) or where two samples follow a given probability distribution. Given a single sample, the distance between the reference probability distribution and the empirical distribution of the given sample is measured. Corresponding to the above distance, a p-value is calculated. Given a significance value $\alpha$. the null hypothesis that the sample follows the reference distribution is acceptable, if the p-value is acceptable if it is greater than the significance value $\alpha$.

In order to compare the attributes of actual prediction and outputs due to adversarial inputs, we consider the error of a TSF model output within a given window. We consider the Root Mean Squared Error (RMSE) between the original prediction with the outputs due to targeted and untargeted attacks. The distribution of the MSE across all the windows in the given dataset roughly follows a normal distribution, making KS-test usable for statistical analysis.

## 4 TARGETED TIME SERIES ATTACKS

Considering the limited number of work in the time series adversarial area and the need for the definition of targeted attacks in the time series domain, we go about the formulation. We take into account the practical aspects that create maximum impact on the output. Depending on the type of impact targeted on the forecasting output, the adversarial attacks are classified as follows:

- **DIRECTIONAL (DTA).** In a Directional Targeted Attack (DTA), the attacker crafts the attack in a way, a minor perturbation on the input causes a shift in the direction of the output. The forecasted output could be shifted upwards or downwards.

- **AMPLITUDINAL (ATA).** In an Amplitudinal Targeted Attack (ATA), the attacker crafts the attack such that a minor perturbation in the input causes the output amplitude to be limited within a prescribed threshold. In such a given scenario, the attacker would want to hide any high-impact areas on the output for his benefit.
- **TEMPORAL (TTA).** In a Temporal Targeted Attack (TTA), the attacker crafts the attack such that a minor perturbation in the input causes a specific time region in the output to be manipulated. In the given region, the attacker would want to change the direction of the output (DTA) or the amplitude of the output (ATA) in a specific time window $(t1, t2)$. The attacker performs DTA or ATA specifically in the target time window to realise TTA. TTA can hence be considered as a sub-category of DTA or ATA.

The attacks are formalized in the context of the respective adversarial attacks (FGSM, PGD, and APGD) in the further sections. In the further equations, the input time series without perturbation is denoted as $x_0$ and the adversarial time series as $x_{adv}$. For amplitudinal attacks (ATA), $\tau$ denotes the threshold to limit the output amplitude. In the case of directional attacks (DTA), the factor $\alpha$ attributes to the direction of the attack and is either +1 or -1 accordingly. For temporal attack (TTA), the type of impact to be created within a given time window $(t1, t2)$ could be either DTA or ATA and is referred as $att(.)$ in the below equations. All the below attacks are defined for $L_\infty$ norm.

## 4.1 Targeted FGSM

**Fast Gradient Sign Method (FGSM).** The FGSM attack Goodfellow et al. (2014a) introduces as notion of gradient-based adversarial attack. The adversarial input 2 is crafted by adding a small amount of perturbation that increases the loss of the true class making the model misclassify the input $x_{adv}$. For a classification example, the perturbation noise is calculated as the gradient of the loss function $\mathcal{L}$ with respect to the input image $x$ for the given true output class. On the other hand, targeted attacks decrease the loss with respect to the target.

$$x_{adv} = x + \epsilon \cdot sign(\nabla_x \mathcal{L}(\theta, x, y_{true}) \tag{2}$$

**Targeted FGSM.** Targeted Attacks for TSF using FGSM are defined as in equation 3, where $\mathcal{L}$ represents the loss or cost function that was used as an optimization function during model training.

$$\begin{aligned} x_{adv,dta} &= x_0 - \epsilon \cdot sign(\nabla_x(\mathcal{L}(\theta, x, y + \alpha \cdot |y|))) \\ x_{adv,ata} &= x_0 - \epsilon \cdot sign(\nabla_x(\mathcal{L}(\theta, x, lim(\tau, y)))) \end{aligned} \tag{3}$$

## 4.2 Targeted PGD

**Projected Gradient Descent (PGD).** The PGD attack is an extension of Iterative-FGSM (popularly called Basic Iterative Method (BIM)) Kurakin et al. (2016). In this iterative method, after each step of perturbation, the adversarial example is projected back into the ball of x using the projection function $\Pi$. The perturbed image after $n$ iterations is denoted by $x_{adv}^n$.

$$x_{adv}^n = \Pi_\epsilon(x^{n-1} + \epsilon \cdot \nabla_x(\mathcal{L}(\theta, x^{n-1}, y_{true}))) \tag{4}$$

**Targeted PGD.** The targeted PGD attacks for TSF extend the targeted attack techniques of FGSM in the PGD context.

$$\begin{aligned} x_{adv}^n &= \Pi_\epsilon(x^{n-1} - \epsilon \cdot \nabla_x(\mathcal{L}(\theta, x^{n-1}, y\prime))) \\ \text{where, } y\prime &\in \{y + \alpha \cdot |y|, lim(\tau, y), att(y(t))\} \end{aligned} \tag{5}$$

## 4.3 Targeted APGD

**Auto Projected Gradient Descent (APGD).** Auto PGD (or simply APGD) is an extension of the PGD attack addressing the sub-optimal step size and objective function issues. APGD Croce & Hein (2020) proposes 1. adding momentum to the gradient step while progressively reducing step size, 2. two conditions to update the step size when there is a successful increase in the objective function and to early-stop when there is no improvement towards the objective function. In algorithm 1, we propose mAPGD-TSF, an extension of the APGD algorithm and is discussed in detail in section:4.3.

---

**Algorithm 1** modified (Targeted) APGD for Time Series Forecasting

---

**Require:** Regression Model $f$, Space $S$, input $x^{(0)}$, step size $\eta$, momentum parameter $\alpha$, iterations $N_{iter}$, checkpoints $W$, target $y\prime$, Loss $\mathcal{L}$

   $x^{(1)} \leftarrow P_S(x^{(0)} - \eta \cdot \nabla \mathcal{L}(f(x^{(0)}), y\prime))$

   **if** $\mathcal{L}(f(x^{(0)}), y\prime)) < \mathcal{L}(f(x^{(1)}), y\prime))$ **then**

      $f_{min} \leftarrow f(x^{(0)})$ and $x_{min} \leftarrow x^{(0)}$

   **else**:

      $f_{min} \leftarrow f(x^{(1)})$ and $x_{min} \leftarrow x^{(1)}$

   **end if**

   **for** $n = 1, N_{iter} - 1$ **do**

      $z^{(n+1)} \leftarrow P_S(x^{(n)} - \eta \cdot \nabla L(f(x^{(n)}), y\prime))$

      $x^{(n+1)} \leftarrow P_S(x^{(n)} + \alpha \cdot (z^{(n+1)} - x^{(n)}) + (1 - \alpha) \cdot (x^{(n)} - x^{(n-1)}))$

      **if** $\mathcal{L}(f(x^{(n+1)}), y\prime)) < \mathcal{L}(f_{min}, y\prime)$ **then**

         $f_{min} \leftarrow f(x^{(n+1)})$ and $x_{min} \leftarrow x^{(n+1)}$

      **end if**

      **if** $n \in W$ **then**

         **if** Condition1 or Condition2 **then**

            $\eta \leftarrow \frac{\eta}{2}$ and $x^{(n+1)} \leftarrow x_{min}$

         **end if**

      **end if**

   **end for**

---

In the targeted APGD attacks, we first go about presenting modified Auto PGD attacks for the Time Series forecasting (mAPGD-TSF) algorithm 1. The modified variant applies to any regression model $f$ in general. The following modifications are made to the algorithm in comparison to AutoPGD to account for targeted time series applications. The changes are namely:

- **Loss function $\mathcal{L}$.** The confidence or the output accuracy is replaced with loss function $\mathcal{L}$. The goal to increase the confidence function is replaced with a goal to minimize the loss with respect to the target.

- **Output Target.** The output target is adapted as per the targeted attacks DTA, ATA and TTA respectively: $y\prime \in \{y + \alpha \cdot |y|, lim(\tau, y), att(y(t))\}$

- **Initialization.** One-step PGD is performed before the start of an iterative reduction of loss. The one-step PGD is considered as an initialization, as the step initializes the minimal loss $f_{min}$.

The conditions related to step size selection are retained from the original AutoPGD Work Croce & Hein (2020). The step size is halved if one of the conditions is true. With $f_{min}^{(n)}$ is the lowest objective value with the least loss in the first $n$-iterations, below are the two conditions:

1. $\sum_{i=w_{j-1}}^{w_j - 1} 1_{\mathcal{L}(f(x^{(i+1)}), y\prime)) < \mathcal{L}(f(x^{(i)}), y\prime))} < \rho \cdot (w_j - w_{j-1})$,

2. $\eta^{(w_j - 1)} \equiv \eta^{(w_j)}$ and $f_{min}^{(w_{j-1})} \equiv f_{min}^{(w_j)}$

## 5 EXPERIMENTS

### 5.1 SETUP: DATA AND MODEL

Energy prediction is a classical problem in time series forecasting. These prediction results need to be accurate however, an attacker could potentially generate adversarial samples in order to alter the predictions which might lead to significant losses for customers and the energy markets. Here we thus study the impact of targeted adversarial attacks on the household electric power consumption data[Repository (2012)]. This dataset comprises measurements of electric power in a household from 2006 to 2010 with a 1-minute sampling rate. In addition to date and time, the dataset contains seven other variables including global active power, global reactive power, voltage, global intensity,

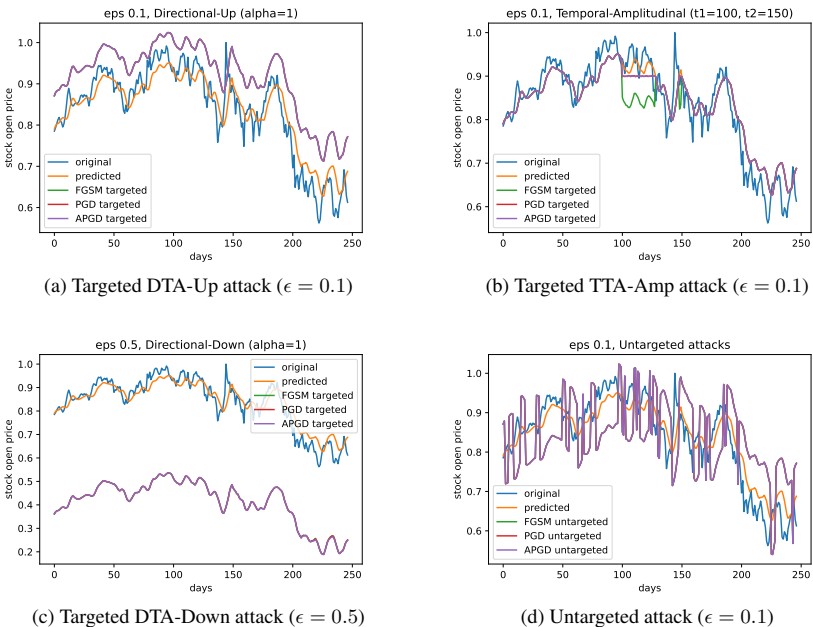

Figure 1: Google stock data prediction on Targeted attacks (a), (b), (c) and untargeted attack (d)

and sub-metering (1 to 3). We use seven variables or input features (global active power, global reactive power, voltage, global intensity, and sub-metering (1 to 3)) to forecast global active power after re-sampling the dataset from minutes to hours on the normalized data.

Stock prediction is a high-demand, high-impact and high-risk area in the financial markets. Stock market prediction can have a direct impact on individuals, organizations, and nation as a whole, thus inaccurate predictions are seen as high-impactful and high-risky. However, these models are also vulnerable to attackers, where one can launch an adversarial attack to make the necessary predictions. Here we study the impact of targeted attacks on the google stock dataset [Dataset (2020)]. This data set comprises google stock prices for the past 5 years i.e., 2017 to 2022 with a 1-day sampling rate (except holidays). In addition to the date, the dataset consists of open, high, low, close, and volume. We use five variables or input features (open, high, low, close, and volume) to forecast the stock open prices on the normalized data.

For both, Google stock and Household electric power consumption datasets, we divide the overall time series into several time windows of size 5. The split for the Google Stock data into train and test datasets with a 4:1 ratio (train data over 4 years: test data over 1 year). For the electrical power consumption data, we split them in a 2:3 ratio (train data over 2 years: test data over 3 years). Different model architectures were experimented for both the datasets including LSTM Hochreiter & Schmidhuber (1997), Vanilla-RNN (Rumelhart & McClelland (1987b), Rumelhart & McClelland (1987a)), GRUCho et al. (2014),and variants of CNN (Hoseinzade & Haratizadeh (2018), Borovykh et al. (2017)). Out of which, LSTM best fits both datasets with minimum RMSE on the test data. Trained the LSTM models using MSE loss with an ADAM optimizer for 50 epochs with early stopping. RMSE on $60\%$ household electric power consumption test data is 0.085, RMSE on $20\%$ google stock test data is 0.040.

## 5.2 TARGETED ATTACKS

We run the proposed targeted attacks i.e. DTA, ATA and TTA using FGSM, PGD, mAPGD for $\epsilon = [0.01, 0.1, 0.5, 1.0, 1.5]$. For DTA, we denote $\alpha = 1$ for direction up, $\alpha = -1$ for direction down for both the datasets. Threshold in ATA, denoted by $\tau$ is taken to be $\tau = 0.9$ for google stock data, $\tau = 0.2$ for household electric power consumption. For TTA, take $(t_1, t_2) = (100, 150)$ and $(t_1, t_2) = (50, 100)$ for both the datasets. We use the MSE loss function in calculating the gradient

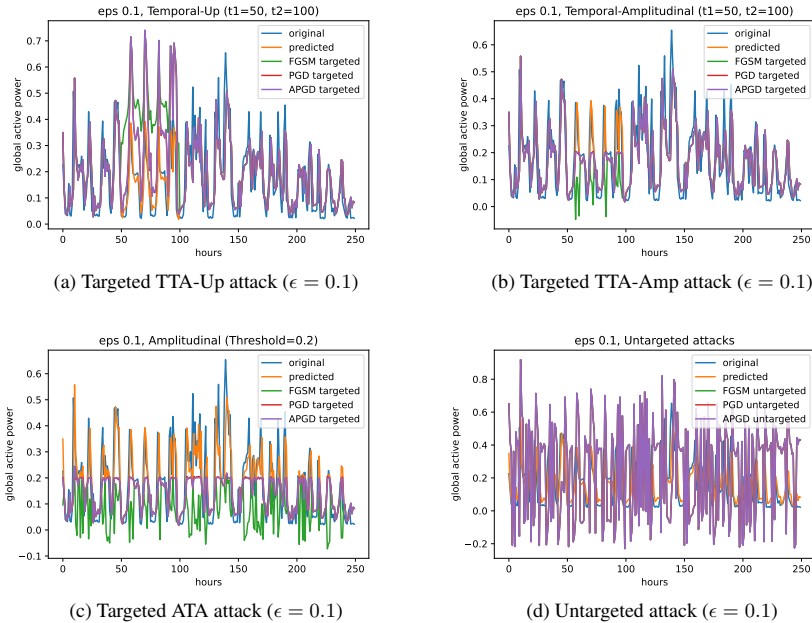

Figure 2: Household electric power consumption data prediction on Targeted attacks (a), (b), (c) and untargeted attack (d)

with respect to inputs. Additionally, to compare the efficacy of the targeted attacks proposed, we run the untargeted attacks using FGSM, PGD, and APGD for different $\epsilon$ values. Untargeted attacks for FGSM, PGD, and APGD follow a notation where the attack results in an increased output loss with respect to the original output. MSE is used as a loss function for gradient calculations as previously done for targeted attacks.

## 6 RESULTS AND DISCUSSION

### 6.1 COMPARISON OF RESULTS

Figure:1 shows the stock open price (original, predicted, FGSM predicted, PGD predicted, APGD predicted) vs across the number of days for targeted attacks with different $\epsilon$ values. We see that the prediction output for FGSM, PDG, and APGD overlap with each other for smaller $\epsilon$-values on DTA and TTA-Up, TTA-Down attacks. For the DTA-Up attack Figure:1(a), the entire predicted output has shifted up compared to the original prediction. Similarly, for the DTA-Down in Figure:1(c), the prediction has moved down. Figure:1(c) also shows the impact of higher epsilon on the output prediction. For TTA-Amp in Figure:1(b), only the prediction between time steps $(t_1, t_2)$ has clipped to the threshold for PGD and APGD. However, here FGSM is not performing as expected. FGSM, instead of clipping the output prediction to the threshold, is bringing the prediction down.

Figure:2 shows the global active power(original, predicted, FGSM predicted, PGD predicted, APGD predicted) vs days for different sorts of targeted attacks. We see that the prediction output for PGD, and APGD overlap with each other, whereas FGSM slightly deviates a bit for DTA and TTA-Up, TTA-Down attacks for smaller $\epsilon$-values. For TTA-Up Figure:2(a), the predicted output between $(t_1, t_2)$ has shifted up compared to the original prediction. Similar case with TTA-Amp in Figure:2(b). Figure:2(c) shows the ATA attack resulting in clipping the output prediction to the threshold. Similar to google stock data, FGSM deviates from PGD, and APGD in ATA attack here as well. On the other hand, untargeted attacks on google stock data Figure:1(d) and household electric power consumption Figure:2(d) results show that there is no control over the prediction output. All three attacks (FGSM, PGD, APGD) increase the loss with respect to true output, but we can't really infer meaningful results from the output prediction since it doesn't have any pattern specified. Hence in

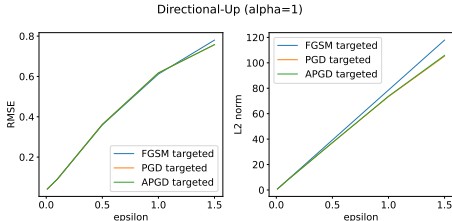
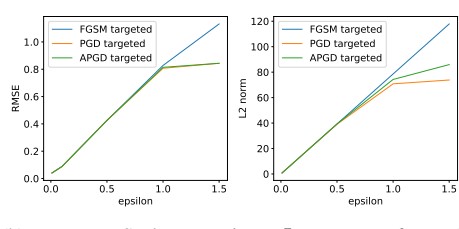

(a) output RMSE loss vs $\epsilon$, input $L_2$ norm vs $\epsilon$ for DTA-Up attack on google data

(b) output RMSE loss vs $\epsilon$, input $L_2$ norm vs $\epsilon$ for DTA-Down attack on google data

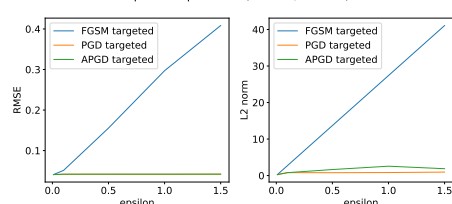
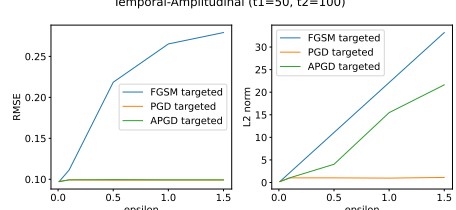

(c) output RMSE loss vs $\epsilon$, input $L_2$ norm vs $\epsilon$ for TTA-Amp attack on google data

(d) output RMSE loss vs $\epsilon$, input $L_2$ norm vs $\epsilon$ for TTA-Amp attack on household electric power consumption data

Figure 3: Loss (RMSE, $L_2$ norm) vs epsilon plots for targeted FGSM, PGD, APGD on google stock data (a), (b), (c) and household electric power consumption data (d)

## 6.2 LOSS VS. AMOUNT OF PERTURBATION

Figure:3 shows the RMSE loss between adversarial output and true output vs epsilon, $L_2$ norm of the adversarial input and true input vs epsilon. We consider $L_2$ norm as the distance metric for analysis since $L_\infty$ can reveal only the maximum perturbation. Results in figure:3(c,d) demonstrate that in ATA attack, and TTA-Amp attack, FGSM has a higher loss in input space as well as in output space compared to PGD and APGD. This justifies the results in figure:1(b), figure:2(b,c) for FGSM not achieving the desired target in the ATA, and TTA-Amp attacks due to the fact that FGSM is single-step attack and performs poorly for targets near the original input (in cases like ATA). On the other hand, PGD and APGD perform well for ATA, and TTA-Amp attacks. PGD and APGD are iterative and successful in reducing the input distance to achieve the attacks. For DTA, TTA-Up, and TTA-Down attacks Figure:3(a,b), FGSM, PGD, and APGD perform similarly for lower values of epsilon. But for higher values of epsilon, FGSM has a higher loss in input space compared to PGD and APGD. Thus, we recommend using PGD and APGD techniques for targeted attacks.

In our experiments, we also studied the effect of the perturbation on the input based on several attributes in the time domain such as the $L_2$ norm. For smaller time windows, the perturbations do not statistically affect the input attributes and hence are immune to any input statistical tests. For larger window sizes, the impact becomes significant but however is not observable in the time domain. It is observed that the perturbations tend to differ from the inputs because of their higher-frequency components. These high-frequency components can then be detected through simple input frequency plausibility tests.

## 6.3 KS TEST RESULTS

We perform statistical analysis on the targeted attacks using the KS-test and analyze the output statistics using histograms. For output statistics, we plot the RMSE loss between the adversarial prediction and the true output for all the targeted attacks in addition to the original RMSE loss, and untargeted RMSE loss in a histogram with a window size of 5 for both the datasets considered.

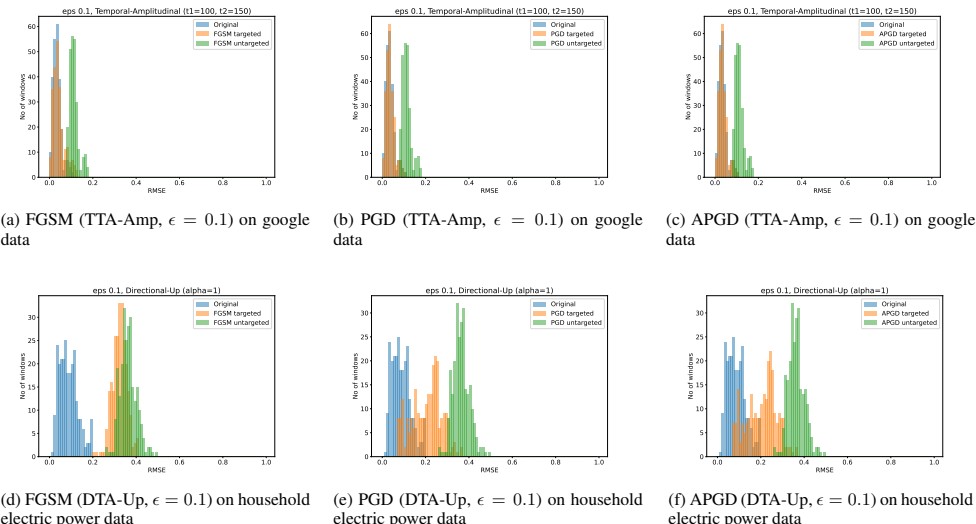

(a) FGSM (TTA-Amp, $\epsilon = 0.1$) on google data

(b) PGD (TTA-Amp, $\epsilon = 0.1$) on google data

(c) APGD (TTA-Amp, $\epsilon = 0.1$) on google data

(d) FGSM (DTA-Up, $\epsilon = 0.1$) on household electric power data

(e) PGD (DTA-Up, $\epsilon = 0.1$) on household electric power data

(f) APGD (DTA-Up, $\epsilon = 0.1$) on household electric power data

Figure 4: Histograms (output statistics) representing the original loss, targeted loss ,and untargeted loss distributions for FGSM, PGD, APGD attacks on the google stock data (a), (b), (c) and household electric power consumption data (d), (e), (f)

| Attack | Google stock | | | | | | | | | Household power | | | | | | | | |
|---|---|---|---|---|---|---|---|---|---|---|---|---|---|---|---|---|---|---|
| | FGSM | | | PGD | | | APGD | | | FGSM | | | PGD | | | APGD | | |
| | O-T | O-U | T-U | O-T | O-U | T-U | O-T | O-U | T-U | O-T | O-U | T-U | O-T | O-U | T-U | O-T | O-U | T-U |
| DTA-up | 0.676 | 0.959 | 0.582 | 0.676 | 0.959 | 0.582 | 0.676 | 0.959 | 0.582 | 1.0 | 1.0 | 0.44 | 0.704 | 1.0 | 0.92 | 0.7 | 1.0 | 0.924 |
| DTA-down | 0.708 | 0.959 | 0.514 | 0.708 | 0.959 | 0.514 | 0.708 | 0.959 | 0.514 | 0.992 | 1.0 | 0.788 | 0.572 | 1.0 | 0.916 | 0.572 | 1.0 | 0.916 |
| ATA | 0.271 | 0.959 | 0.769 | 0.113 | 0.959 | 0.959 | 0.113 | 0.959 | 0.959 | 0.436 | 1.0 | 0.936 | 0.12 | 1.0 | 0.984 | 0.116 | 1.0 | 0.984 |
| TTA-up | 0.137 | 0.959 | 0.931 | 0.137 | 0.959 | 0.931 | 0.137 | 0.959 | 0.931 | 0.208 | 1.0 | 0.848 | 0.152 | 1.0 | 0.968 | 0.152 | 1.0 | 0.968 |
| TTA-down | 0.157 | 0.959 | 0.858 | 0.157 | 0.959 | 0.858 | 0.157 | 0.959 | 0.858 | 0.196 | 1.0 | 0.94 | 0.14 | 1.0 | 0.988 | 0.14 | 1.0 | 0.988 |
| TTA-amp | 0.117 | 0.959 | 0.882 | 0.044 | 0.959 | 0.959 | 0.044 | 0.959 | 0.959 | 0.084 | 1.0 | 1.0 | 0.036 | 1.0 | 1.0 | 0.036 | 1.0 | 1.0 |

Table 1: KS test (Output statistics) for all the proposed targeted attacks based on comparing the original loss, targeted loss, untargeted loss distributions on google, household electric power consumption dataset for $\epsilon = 0.1$. Here O-T, O-U, T-U represent the KS statistics between Original-Targeted, Original-Untargeted, and Targeted-Untargeted attacks. The lesser the values imply the closer the two distributions.

Figure:4 demonstrates that PGD and APGD targeted are statistically similar to the original compared to FGSM targeted.

Table:1 shows the statistical difference between original, targeted, and untargeted attacks for $\epsilon = 0.1$. Table:1 indicates that in google stock data, FGSM is statistically far from original compared to PDG, and APGD in ATA attack. Also for household electric power consumption data, FGSM is far from the original compared to PGD and APGD in almost all the attacks. It is observable from the table:1 and figure:4 that i) targeted attacks are more statistically similar to the original prediction compared to untargeted attacks, this infers that targeted attacks are less likely to be detectable compared to untargeted attacks. ii) targeted PGD and APGD perform better compared to FGSM.

## 7 CONCLUSION

In this paper, we introduced the first notion of targeted attacks on Time Series Forecasting. We define and formalize three types of targeted attacks using existing white box attack techniques. Together with FGSM and PGD, we proposed a modified version of AutoPGD attacks for regression tasks to apply the formulation. Our experimental results prove successful targeted attacks on the forecasting outputs. Through histograms and KS-test, we statistically show that the targeted attacks are closer and hence undetectable compared to untargeted attacks. Further work in the directions includes black-box extensions and analysis of input statistics in the transformed frequency domain. This work opens up a new direction toward the exploration of defenses for TSF.

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
