# OpenReview forum: "Targeted Attacks on Timeseries Forecasting"
_ICLR.cc/2023/Conference — Submitted to ICLR 2023_

### Official Review · Reviewer_6F4s · 2022-10-16

**Confidence:** 5
**Correctness:** 2
**Technical Novelty And Significance:** 2
**Empirical Novelty And Significance:** 1
**Recommendation:** 3

**Clarity, Quality, Novelty And Reproducibility:**

Overall, the presentation of the paper is below the standards of quality for ICLR.

The quality of the English text is poor (but the manuscript is readable).

Figures and Tables are (aesthetically) fair.

The topic addressed by the manuscript is trendy, and important.

The references are (somewhat) appropriate.

The contribution is not significant (currently).

The experiments are partially reproducible (the data is public and the algorithms are described in the text, but the code is not disclosed).

The novelty is poor: attacks against ML-based methods for TSF have been proposed in the past, and the provided formalization is too vague to determine whether it is truly a novel contribution or not.

**Strength And Weaknesses:**

STRENGTHS:
+ Trendy topic that has been not very explored
+ The statistical validation is interesting


WEAKNESSES
- Very poor presentation
- Lack of scientific rigour (in clarity and quality)
- Unrealistic assumptions
- Biased experiments


**Summary Of The Paper:**

The paper tackles the problem of adversarial attacks against Machine Learning (ML) methods for Time Series Forecasting (TSF). The paper rightly points out that there are few works in this domain: hence, the paper proposes an original formalization of “targeted attacks” against such methods, which also works for "regression" ML problems (typical in TSF, but rare in the 'traditional' computer vision setting envisioned in adversarial ML research). The proposed attacks are then empirically validated on two datasets (representing two diverse real-world settings) and the effectiveness of the attacks is validated via statistical tests.

**Summary Of The Review:**

First, I appreciate the line of research tackled by this paper. It is true that there have been few papers that investigated “adversarial attacks against ML methods for time series forecasting”. Furthermore, it is also true that these methods are becoming increasingly popular in the real world, as they are being integrated into production systems. Therefore, it is also crucial to study this subject from a security viewpoint.

Unfortunately I cannot recommend acceptance of this paper – at least not in its state, and definitely not for ICLR. Indeed, the paper is affected by several and critical weaknesses, which affect both its “security” focus and its technical rigour (required for ICLR). I will elaborate below on the many problems that affect the paper, and provide suggestions for improvement.

Before I do this, however, I must warn the authors that I will gladly engage in a discussion with them, but that I am also very unlikely to change my mind on the submission, as I believe the amount of effort required to put it into an “acceptable” state for ICLR is simply too much to address during the rebuttal phase.

**Limited scope.**
According to Section 3.1, the considered setting assumes the usage of Time Series Forecasting only for predicting the “next” value of a given time-series. Such a setting can be a bit simplistic as there are several cases in which the forecasting may entail the prediction of “multiple” future values. Albeit one can argue that predicting multiple values can be done by iteratively predicting a single one, it is unclear whether this scenario also falls within the threat model of the paper.

**Vague definitions.**
The following “definition” of adversarial time series is not rigorous: “The adversarial time series $\hat{X}$ ($X_{adv}$) is intended to significantly worsen the output prediction $\hat{Y}$ of a TSF model.” Given the scope of the paper (let me quote from the introduction: “We define and formalize targeted attacks on deep learning time series forecasting.”), such “vagueness” is unacceptable. The same issue affects the definition of the attacker’s “goal”, which I quote: “The goal of the attacker is to create a targeted output impact on the time series.” What is a “targeted output impact”? Furthermore, I do not agree with this statement: “Properties of the perturbation. The performance of the model prediction would reduce as the strength of the perturbation increases.” It may be intuitive to assume that “larger” perturbations have a greater impact, but such a statement is not universal. Do the authors *require* their perturbations to have a stronger impact as their magnitude is increased? Finally, the following sentence is problematic: “This attributes to the cost of data manipulation. It is notable that it is more expensive to achieve higher input perturbation”: first, there is no mentioning of this “cost”, thereby increasing the vagueness; second, because “greater perturbation = greater cost” is also not universally true (see [A]). Put simply, the authors attempt to formalize adversarial attacks on TSF, but the provided definitions are vague and the formalization – which appears to be universal – is limited to very few specific cases (which are not even properly defined).

**Extremely Powerful Attacker.**
The proposed threat model envisions an extremely powerful attacker, who has white-box knowledge of the targeted ML model. This is highly unrealistic in the real world; furthermore, although the authors claim that “the attacks work even in black-box settings”, the way in which such attacks work is by iteratively querying the targeted ML model---thereby assuming an attacker who has “oracle access” to the ML model. Both scenarios are highly unlikely in real world settings (see, e.g., [A]), especially because the attacker – on top of knowing/being able to query the targeted ML model – also needs to introduce the perturbations (i.e., they must be able to manipulate the input data stream analyzed by the ML model). Simply put, from a security standpoint, the considered “threat model” is unrealistic and I am not surprised in the slightest that the attacks “work”. I invite the authors to provide some specific use cases of the proposed threat model, which should be provided with concrete justifications as to why an attacker would invest so much effort in setting-up, and then launching, a similar offense. Otherwise the authors should remove these considerations and simply state that the proposed “contribution” is devoted to assessing the robustness of ML models for TSF to adversarial examples (which are not necessarily malicious). I point the authors to a recent work that summarizes all these issues, [A]


**The definition of “temporal targeted attacks” (TTA) is flawed.**
According to Section 4, these attacks are defined as follows: “In a Temporal Targeted Attack (TTA), the attacker crafts the attack such that a minor perturbation in the input causes a specific time region in the output to be manipulated. In the given region, the attacker would want to change the direction of the output (DTA) or the amplitude of the output (ATA) in a specific time window $(t_1, t_2)$.”. This definition is in stark contrast with the definition of “time series forecasting” provided in Section 3, which I quote: “Given a time series X = [$x_1$, $x_2$, … $x_T$], a time series forecasting task predicts the value of $x_{T+1}$ based on the previous samples [$x_{T−w}, x_{T−w+1}, ..., x_{T}$], where $w$ is the window size under consideration.” Indeed, if the goal of time series forecasting is predicting $x_{T+1}$, then how can the attacker influence the output in a specific time window $(t_1, t_2)$? This is only possible if $t_1=t_2$, which means that the window is not a window in the first place. Therefore, either one of the following is true: the definition of time series forecasting is flawed, or TTA are redundant.


**No (realistic) justification for the Experimental Evaluation.**
This is a follow-up issue to the “extremely powerful attacker” described above. Let me quote from Section 5: “Stock prediction is a high-demand, high-impact and high-risk area in the financial markets. Stock market prediction can have a direct impact on individuals, organizations, and nation as a whole, thus inaccurate predictions are seen as high-impactful and high-risky. However, these models are also vulnerable to attackers, where one can launch an adversarial attack to make the necessary predictions.” Specifically, “how can an attacker launch an adversarial attack against ML models tasked to predict stock market values?” According to the above, the attacker must be able to manipulate the data, and either know the targeted ML model (which is an IP of a company), or be able to query such ML model. How is this realistically feasible? Reality is far more complex and my impression is that the paper is being oversold (from a security perspective), thereby increasing the confusion (endemic among practitioners) about the real threat of adversarial attacks against ML models [A]. (Note that the same comment applies also for the “energy prediction” scenario: why, and how, would an attacker attempt to do so?)

**Biased experiments.**
For the evaluation, the authors adopt some parameter settings without properly motivating them. Why was the Google Stock Prices split into train:test with a 4:1 split? Why were the time windows set to 5? Maybe there is a logic, but as it is not described in the paper, I am inclined to believe that the authors arbitrarily “cherry picked” the values that confirmed their attacks to be successful.

**Unclear statistical validation.**
It is unclear how the KS test was carried out. Table 1 shows the resulting p-value of the many comparisons performed in the paper. However, it is not clear what was actually being compared: did the authors simply aggregate the overall results obtained by a (single) ML model on the the (single) train:test split with a (single) parameter configuration? In other words, how large is the size of the “populations” that are being compared? My guess is that the authors performed such tests by considering a single “configuration”, which is not fair and definitely not enough to support any claim about the effectiveness of the attacks. I invite the authors to consider diverse configurations (e.g., different splits, different window sizes), and then use all such diverse configurations to perform a (more transparent) statistical test by also adopting appropriate “correction” techniques (e.g., Bonferroni Correction). Doing so would tremendously help supporting the effectiveness claims made in the paper!

Some additional issues:

•	There is no buildup for Section 3.2.

•	This sentence makes no sense (Section 3.2): “Corresponding to the above distance, a p-value is calculated. Given a significance value α. the null hypothesis that the sample follows the reference distribution is acceptable, if the p-value is acceptable if it is greater than the significance value α.”

•	The quality of writing in Section 4 is very low

•	I am inclined to believe that the proposed “ATA” are just a subset of “DTA”. A more rigorous definition is necessary

•	The following statement is wrong: “The FGSM attack Goodfellow et al. (2014a) forms the basis of any known adversarial attack techniques.” There are a plethora of “adversarial attacks” wherein the adversarial examples are created by means of techniques that have nothing to do with FGSM. I invite the authors to avoid making such statements.

•	The paper should make an effort to cite some existing works that proposed adversarial attacks against TSF for the two considered use cases (a quick search on Google Scholar would reveal several papers that evaluated their attacks on such data).

•	The proposed attack is evaluated only on "univariate" time series. Will it work also on multivariate time series? Why has this setting not been considered in the paper?


EXTERNAL REFERENCES

[A]: Apruzzese, G., Laskov, P., de Oca, E. M., Mallouli, W., Rapa, L. B., Grammatopoulos, A. V., & Franco, F. D. (2022). The Role of Machine Learning in Cybersecurity. Digital Threats: Research and Practice.

---

> ### Author Response · Authors · 2022-11-18
> **Response to Reviewer 6F4s**
>
> Dear Reviewer,
>
> Thank you for the review. We would like to appreciate your acknowledgement that the topic is less explored. The constructive comments have helped us to improve several sections and the quality of the paper. We would like to address your concerns individually below:
>
> Q1: Limited Scope:
>
> A1: We have considered the uni-variate use case that predicts the next value of the given time series. Albeit limited, we have chosen this setting based on - focus on the number of previous work in time series, number of practical usecases for univariate and single "next" prediction. However, as you have rightly pointed, we could further extend this work by iteratively predicting the values.
>
> Q2: Vague Definitions
>
> A2: The definitions are more generalized as the output impact could be different for different tasks. For example, the attacker on the stock prediction would want to increase the prediction if he/she was to sell and decrease the prediction if he/she was to invest. Similarly for the housing dataset, the attacker as a consumer of electricity would want to decrease the output electricity consumption. We have made attempts in the rebuttal version to provide more clarity on these points.
>
> Q3: Extremely Powerful Attacker
>
> A3: The threat model considers a (relatively) powerful attacker in a practical setup. We base our threat model on the following prior works. The work by Yuan, Xiaoyong et al.[1] back in 2018 provides the overview and possibility of the attacks. The work by Szegedy et al.[2] and Papernot et al.[3] propose how adversarial examples generated by one neural network can be used to attack another neural network. In most cases, the attacker has some knowledge about the possible model architecture, the task for which it is used, data available from open sources or have access to the system hosting the model itself (Embedded devices for example, leading to possibilities of Hardware Attacks). In cases where the system could possibly restrict the number of queries from a potential attacker, the attack could query as though he was more than one user and use the cumulative query information to strengthen the surrogate model. Further, the combination of one or more technique significantly improves the attackers capabilities. The work [A] is very interesting and considers several situations of Machine Learning for Cybersecurity. The paradigm of using Machine learning for cybersecurity for an attacker opens new attack surfaces, which would make the attacker more powerful.
>
> Q4: The definition of “temporal targeted attacks” (TTA) is flawed.
>
> A4: We highly appreciate the comment, it has helped us to reorder the definitions for better understanding of the reader. As you have rightly pointed, the definition of time-series is formulated for the single time step prediction $x_{T+1}$ based on the previous samples. The notion of (attack) window is the number of time steps where the attacker wants to create the attack (either directional or amplitudinal). The Temporal-Targeted Attacks, as already stated in the equations, are DTA or ATA in specific time window. We have reconsidered to take TTA as sub-categories under DTA and ATA for more clarity.
>
> Q5: No justification for the Experimental Evaluation:
>
> A5: We have considered the statistical justification to the experimental evaluation with the setup as defined in the threat model. This assumes that the attacker is able to access the model and hence other aspects of the attacker kill-chain (traditional cybersecurity relevant) are not discussed in detail.
>
> Q6: Biased experiments:
>
> A6: The choice of dataset as Google stock data and Electricity consumption data was intended as it is the most commonly used datasets for analysis of time series forecasting [4,5,6]. We also chose a similar data split from these works. Due to the number of sample of the dataset, we could not analyse the outcome for large window sizes (in 100s or 1000s). Our experiments indeed includes different window sizes (in 10s). The attack efficacy varies for the different window sizes, which could be used a comparison.
>
> Q7: Unclear statistical validation:
>
> A7: At this point, we consider a single ML model with a single train:test split, as the intention is largely formalize and examine the applicability of the proposed method across other known attack techniques (FGSM, PGD and APGD).
>
> References:
> [1] Yuan, Xiaoyong et al, 2019 “Adversarial Examples: Attacks and Defenses for Deep Learning.” ;
> [2] C. Szegedy et al, 2013 “Intriguing properties of neural networks,” ;
> [3] Papernot et al,  “Transferability in machine learning: from phenomena to black-box attacks using adversarial samples ;
> [4] Fawaz et al, 2019.  Adversarial attacks on deep neural networks for time series classification. ;
> [5]  Tao Wu et al, 2022. Small perturbations are enough: Adversarial attacks on time series prediction. ;
> [6] Dang-Nhu et. al, ICML '20. Adversarial attacks on probabilistic autoregressive forecasting models.;

---

### Official Review · Reviewer_zM1v · 2022-10-21

**Confidence:** 5
**Correctness:** 2
**Technical Novelty And Significance:** 2
**Empirical Novelty And Significance:** 2
**Recommendation:** 3

**Clarity, Quality, Novelty And Reproducibility:**

The submission completely overlooks the related work of Dang-Nhu et al. [1] which performs adversarial attacks on the well known DeepAR model, applied to stock price prediction and electricity consumption forecasting. Dang-Nhu et al. present:
- a general setting with arbitrary target function
- a threat model that is similar to the submission
- a probabilistic framework that allows to perturb the whole output distribution
- an iterative setting that permits application to multi-step forecasting
- a detailed discussion on the attack's gradient estimators

This seems to be more a much more general approach than what is discussed in the submission, dating from 2020. There are certainly other related papers that are not cited.

[1] Raphaël Dang-Nhu, Gagandeep Singh, Pavol Bielik, and Martin Vechev. Adversarial attacks on probabilistic autoregressive forecasting models, ICML 2020.


**Strength And Weaknesses:**

The paper is clear and the experiments appear sound. The use of proper statistical tests is appreciated. However, I have serious concerns about the originality and novelty of the contribution, because the paper overlooks directly related work on targeted attacks for time series forecasting. Besides, Section 4 (4 -Targeted Time Series Attacks) looks more like a review of existing attacks rather than a proper contribution, though I acknowledge that the APGD algorithm has been adapted.

**Summary Of The Paper:**

The paper extends targeted adversarial attacks to time series forecasting, and performs a statistical evaluation to demonstrate the effectiveness of the methods.

**Summary Of The Review:**

Because of serious concerns about novelty and originality, I can not recommend acceptance.

---

> ### Author Response · Authors · 2022-11-18
> **Response to Reviewer zM1v**
>
> Dear Reviewer,
>
> Thank you for your time and response for the review and the comments. We would like to address the concerns below:
>
> Q:"The submission completely overlooks the related work of Dang-Nhu et al"
>
> A: Thank you for the comment, we will elaborate the differences in the rebuttal version to bring out more clarity. The work from Dang-Nhu et al considers a probabilistic framework and more specifically works towards perturbation on the whole output distribution. In our paper, we relax the constraint to apply the proposed method to any (probabilistic or non-probabilistic) deep learning model. Additionally, the proposed method can be adapted to any gradient based method. In this paper, we show that the method can be used with FGSM, PGD and APGD. Further, the proposed method can be applied to any model where only the output is available and not the whole output probability distribution. In the considered practical setting, the output probability is not always available.
>
> In the Threat Model setting, we consider a two-step approach where the attacker firstly obtains a surrogate model. Through the transferable property, the attacker is able to use the surrogate model (as a white box) to craft the adversarial for the victim model to be attacked. Hence the gradient estimation is dependent on the approximation of the surrogate model to that of the victim model.

---

### Official Review · Reviewer_oRvB · 2022-10-23

**Confidence:** 3
**Correctness:** 4
**Technical Novelty And Significance:** 2
**Empirical Novelty And Significance:** 2
**Recommendation:** 5

**Clarity, Quality, Novelty And Reproducibility:**

The main observation of the paper appears to be that targeted attacks are more effective than untargeted attacks, which must hold quite generally and should not be limited to this particular question. So, while the paper is understandable and clear, it is not clear what the novelty is.

**Strength And Weaknesses:**

The paper nicely summarizes various popular attack patterns for deep learning methods and uses them in a time series forecasting setup. Notably absent from the paper is a meaningful defense mechanism to a targeted attack.

**Summary Of The Paper:**

The authors propose to study targeted attacks on deep learning methods for time series forecasting. They adapt existing approaches to attack trained neural networks for image classification and run a number of experiments.

**Summary Of The Review:**

The paper appears to apply established techniques to a related problem. This contribution appears to be marginal in terms of novelty.

---

> ### Author Response · Authors · 2022-11-18
> **Response to Reviewer oRvB**
>
> Dear Reviewer,
>
> We thank you for the review and your comments. We would like to provide clarity specifically on the point on novelty and the effectiveness of the attack.
>
> In the paper, we argue that the targeted attacks are effective than that of untargeted attacks using the outputs loss distributions (Distribution of the forecast errors) for statistical comparison. The similarity of the error distribution of the targeted attacks to the distribution of error of the original signal makes the targeted attacks less prone to detection against statistical (post-processing) detection mechanisms.
> The method proposed in this paper is applicable to any known gradient based attacks. we have considered the attacks - FGSM, PGD and APGD to discuss the performance of the attacks when using these techniques.
>
> Proposal of Defense: We limit the scope of work to the attacks and propose the work on defense as future work and hence not covered as part of this paper. We will use the hints provided into the future directions of research.
>
> We would be happy to address if there are any specific questions as well.

---

### Official Review · Reviewer_X1J3 · 2022-10-24

**Confidence:** 3
**Correctness:** 4
**Technical Novelty And Significance:** 3
**Empirical Novelty And Significance:** 2
**Recommendation:** 5

**Clarity, Quality, Novelty And Reproducibility:**

**Clarity**: The paper is well-written and easy to read.

**Quality, Novelty:** This paper proposes a novel targeted attack on the TFC task. However, adding more details to describe the attack methods more clearly will be great. The evaluation cannot fully support the effectiveness of the adversarial attack.

**Reproducibility**: The paper provides some experiment setup but does not provide the code.

**Strength And Weaknesses:**

Strength:

+ Clean writing: The paper is well-written and easy to read.

+ Novel method: The paper proposes targeted adversarial attacks against time-series forecasting tasks based on three different attack goals: directional attack, amplitudinal attack, and temporal attack.

+ Empirical experiment support: The paper evaluates the proposed methods on two time-series datasets and shows that the proposed attack method achieves the attack goals.

Weakness:

+ Lack of methodology details: I think the most interesting part of the paper is the different targeted attack goals and their corresponding strategies to achieve these goals. In equation 3, amplitudinal and temporal attack needs $lim$ and $att$. Although there is a discussion before Section 4.1 on these two functions, I am still not quite clear about these functions are defined. It will be great to describe these two functions more formally to avoid unnecessary confusion.

+ The effectiveness of the attack: In Figure1, if we compare the delta between the original, predicted, and attacked predicted values, we can find that the delta is always less than $0.1$. Given the $\epsilon=0.1$, a $0.1$ change in predicted values can be considered expected. For example, for a DTA-up attack, if we simply increase the input value by $0.1$, we may also be able to increase the predicted value. This change makes me feel the adversarial attack is not effective enough. It will be good to visualize the perturbations and the adversarial examples so that readers can better understand how adversarial perturbations cause unexpected changes in the forecasting results.

+ Also, where are the red lines (PGD) and green lines (FGSM) in Figure 1? And why is only the DTA-down attack with $\epsilon = 0.5$? What’s the performance of the DTA-down attack with $\epsilon =0.1$? Based on the value on the y-axis, $0.5$ is a very large perturbation.


**Summary Of The Paper:**

This paper studies the targeted attack on time-series forecasting tasks. Specifically, this paper proposes three different targeted attack goals: directional attack, amplitudinal attack, and temporal attack. This paper then designs the corresponding FGSM, PGD, and APGD attacks for these goals. All attacks are evaluated on the google stock dataset and household electric power dataset, and the evaluation results show that the proposed attacks achieve adversarial attack goals.

**Summary Of The Review:**

Thanks for the paper, I thoroughly enjoyed reading the paper. My concern about the paper is the effectiveness of the targeted attacks.

---

> ### Author Response · Authors · 2022-11-18
> **Response to Reviewer X1J3**
>
> Dear Reviewer,
>
> We thank you for the constructive comments and the suggestions. We appreciate your comment that the paper is easy to read. We would like to provide clarity on the concerns as follows:
>
> Q1: Lack of Methodology Details:
> A1: Your comment is highly appreciated and we will add specific parts to equations in section 4.1. Moreover, based on the comments from other reviewers, we would like to classify the temporal attack as subset of directional and amplitudinal attacks to bring more clarity in temporal attack formulation and equations.
>
> Q2: Effectiveness of the Attack:
> A2: Visualizing the input changes due to adversarial perturbations would be a good way to give more clarity on the effectiveness of the attack. We wish to add these additional visualizations in the Appendix section. Through the depiction, we also want to clarify that the perturbation of 0.1 is the maximum change on the input for a single input feature.
>
> Q3: Details in the Figures:
> A3: Some of the lines are not visible as they overlap with each other. The reason being, for lower $\epsilon$-values, PGD, FGSM and APGD produce identical effects on the outputs (in this case output RMSE). We explain this in section 6.2 and these are also evident from figure 3. We will highlight this to provide more clarity to the reader.

---

> > ### Comment · Reviewer_X1J3 · 2022-11-22
> > **Thanks for the response.**
> >
> > I really appreciate the authors’ response and extra clarification.
> >
> > I don’t think the answer to Q2 addresses my major concern:
> > Given the $\epsilon=0.1$, a $0.1$ change in predicted values can be considered expected: the perturbation of 0.1 is the maximum change on the input for a single input feature, but the output only changes about 0.1. In other words, if we simply minus 0.1 on all input features (which is not an adversarial attack), the output features can also have similar changes. If the attack is effective enough, I expect a more significant change (at least significantly larger than 0.1).

---

> > > ### Author Response · Authors · 2022-12-12
> > > **Additional response to Reviewer X1J3 on Perturbation strength**
> > >
> > > Thank you for the specific question and the clarification.
> > >
> > > In case of DTA, for the two datasets used in the experiments, we have seen that the perturbation with strength '$\epsilon$' leads to an output difference similar to the '$\epsilon$' considered. Hence, we see a change of $0.1$ corresponding to the $\epsilon = 0.1$. The maximum change of $0.1$ could be added to the input feature based on the gradient calculation. If the gradient is $0$, then there is no change in that input feature.
> > >
> > > In general, we see that the effect of the input features strength on the output is not always one-to-one. When multiple input features are involved, some features contribute significantly compared to others. Adding perturbations to these significant (important) features result in significant output changes (especially notable in DTA).
> > >
> > > We appreciate the hint as it drives us to study how feature importance could play a role in the input perturbations. Please let us know if we could provide more clarity.

---

### Decision · Program_Chairs · 2023-01-20

**Decision:**

Reject

**Justification For Why Not Higher Score:**

The work has potential but its current form is far from the acceptance threshold.


**Justification For Why Not Lower Score:**

N/A


**Metareview: Summary, Strengths And Weaknesses:**

The study of adversarial attacks on time series forecasting models is much less studied than those on other machine learning models, as correctly pointed our by the authors. As such, this work is well positioned on an important topic. Some specific methods in the experiments such as the statistical validation are also interesting. Nevertheless, the work in its current form is still very premature for publication. Major concerns include insufficient novelty, lack of scientific rigor, and unsatisfactory presentation, making it far below the acceptance level of ICLR. We hope the authors find our comments and suggestions useful to revise their paper for future resubmission.